# Self-Reporting Theranostic: Nano Tool for Arterial Thrombosis

**DOI:** 10.3390/bioengineering10091020

**Published:** 2023-08-29

**Authors:** Suryyani Deb, Mohammad Azharuddin, Sofia Ramström, Kanjaksha Ghosh, Santiswarup Singha, Thobias Romu, Hirak Kumar Patra

**Affiliations:** 1Department of Biotechnology, Maulana Abul Kalam Azad University of Technology, Haringhata 741249, India; 2Cambridge Display Technology Ltd., Huntingdon PE29 2XG, UK; 3School of Medical Sciences, Orebro University, 701 82 Örebro, Sweden; 4National Institute of Immunohaematology, KEM Hospital Campus, Mumbai 400012, India; 5National Institute of Immunology, Aruna Asaf Ali Marg, New Delhi 110067, India; 6Centre for Medical Image Science and Visualization (CMIV), Department of Biomedical Engineering (IMT), Linköping University, 581 85 Linköping, Sweden; 7Department of Surgical Biotechnology, UCL Division of Surgery and Interventional Sciences, University College London, London NW3 2PF, UK

**Keywords:** arterial thrombosis, theranostics, platelet aggregations, MRI, ReoPro

## Abstract

Arterial thrombosis (AT) originates through platelet-mediated thrombus formation in the blood vessel and can lead to heart attack, stroke, and peripheral vascular diseases. Restricting the thrombus growth and its simultaneous monitoring by visualisation is an unmet clinical need for a better AT prognosis. As a proof-of-concept, we have engineered a nanoparticle-based theranostic (combined therapy and monitoring) platform that has the potential to monitor and restrain the growth of a thrombus concurrently. The theranostic nanotool is fabricated using biocompatible super-paramagnetic iron oxide nanoparticles (SPIONs) as a core module tethered with the anti-platelet agent Abciximab (ReoPro) on its surface. Our in vitro feasibility results indicate that ReoPro-conjugated SPIONS (Tx@ReoPro) can effectively prevent thrombus growth by inhibiting fibrinogen receptors (GPIIbIIIa) on the platelet surface, and simultaneously, it can also be visible through non-invasive magnetic resonance imaging (MRI) for potential reporting of the real-time thrombus status.

## 1. Introduction

Atherosclerosis is a progressive disease characterised by the accumulation of lipids, fibrous materials, and minerals in the arteries, leading to the formation of plaque. Plaque disruption exposes thrombogenic matrix proteins, e.g., von Willebrand factor (VWF), fibrin, fibrinogen, thrombospondin, vitronectin, fibronectin, and collagens, etc., which can recruit and activate circulating platelets. This leads to arterial thrombosis (AT) and may block arteries and cause blood circulation impairment [1]. AT contributes to cardiovascular diseases (myocardial infarction and ischemic stroke) upon plaque disruption [2,3]. Globally, AT is one of the leading causes of morbidity and mortality [4]. Recruitment of platelets in the plaque rupture site initiates thrombus formation and can obstruct blood flow [5]. Due to this fact, platelets become an important target to prevent arterial blockage during plaque disruption in atherosclerosis [6].

Currently, diagnosis of AT relies on several techniques, such as X-ray angiography, magnetic resonance angiography (MRA), or coronary computerised tomography angiography (CTA). However, most of them are highly invasive [7]. Furthermore, a potential risk is associated with unwanted plaque rupture during measurement, as in the case of angiographic techniques [8]. Hence, a suitable therapy is immediately recommended, followed by a detection. One of the most sensitive and reliable non-invasive diagnostic techniques is cardiac magnetic resonance imaging (CMRI). CMRI is a painless, accurate, and precise way to identify high-risk atherosclerotic plaque [9,10]. CMRI has been well-accepted among clinicians irrespective of the risks associated with the contrast agents used. Gadolinium is the most commonly used contrast agent for MRI but is associated with severe toxic effects as it cannot be metabolised in the body and is retained long after completion of the MRI scan [11]. Super-Paramagnetic Iron Oxide Nanoparticles (SPIONs) are emerging as a potential alternative to minimise such toxic side effects owing to their promising metabolic profiles inside the body [12], along with their high spatial resolution in magnetic imaging. We have rationally chosen SPIONs because of their safety index and clinical use. FDA has already approved several iron oxide nanoparticle-based systems, such as Venofer^®^, Ferrlecit^®^, INFed^®^, Dexferrum^®^, and Feraheme^®^, etc., to name a few. However, similar to other nanosystems, bare SPIONs are potential activators of platelets in specific conditions [13]. Thus, thrombus detection using safer SPION conjugates to avoid cardiovascular events is very much needed.

In this proof-of-concept study, we have described the development of a bio-compatible, SPION-based theranostic nanosystem (Tx@ReoPro) with dual properties (i) to act as a therapeutic agent against thrombus growth and (ii) to monitor the growth restriction. We have confirmed that Tx@ReoPro does not exhibit any pro-aggregatory effect on platelets but rather restricts thrombus growth effectively by inhibiting platelet aggregation. Our synthesised Tx@ReoPro comprises the model companion drug ReoPro (or Abciximab, a clinically approved antiplatelet drug that has been prescribed during percutaneous coronary interventions (PCI). ReoPro has the potential to regulate thrombus growth, and MRI-sensitive SPIONs function as contrast agents for MRI for real-time disease surveillance. To the best of our knowledge, this is the first-ever feasibility study reporting on a theranostic approach that includes both AT detection and therapy concurrently in an in vitro set-up.

## 2. Materials and Methods

### 2.1. SPION Synthesis by Thermal Decomposition Method

SPIONs were synthesised by the previously reported thermal decomposition method [14]. Briefly, 1.667 g of iron acetylacetonate (Sigma-Aldrich, Saint Louis, MO, USA) was mixed with 25 mL benzyl ether (Sigma-Aldrich, Saint Louis, MO, USA) and 25 mL oleylamine (Sigma-Aldrich, Saint Louis, MO, USA). The mixture was then heated at 110 °C in the presence of nitrogen purging, which afterwards was increased to 310 °C and continued for 2 hrs in reflux condition in a rotating heating mantle. After cooling down to room temperature, 60 mL ethanol was added to it. The final suspension was then centrifuged at 3000× *g* for 20 min, and the supernatant was discarded. The precipitate was allowed to air dry and was finally resuspended in 20 mL hexane (Sigma-Aldrich, Saint Louis, MO, USA). The hexane suspension was then centrifuged at 25,000× *g* for 1 hr to pellet the larger particles, and the supernatant was collected for further modification.

### 2.2. Bio-Conjugation of Iron Nanoparticles

An amount of 200 mg polyethylene glycol dicarboxylic acid (HOOC-PEG-COOH, 2kd) (JenKem Technology, Allen TX, USA), 9 mL di-methyl formamide (DMF) (Sigma-Aldrich, Saint Louis, MO, USA), and 20 mL chloroform (Sigma-Aldrich, Saint Louis, MO, USA) were mixed in a reaction flask. Then, 50 mg NHS (N-Hydroxysuccinimide) and 50 mg EDC (1-Ethyl-3-(3-dimethylaminopropyl) carbodiimide, (both from Thermo Fisher Scientific, Waltham, MA, USA) were dissolved separately in 330 μL di-methyl formamide (DMF) and added to the reaction flask. The mixture was then stirred at room temperature for 30 min. To that, 11 mg dopamine (Sigma-Aldrich, Saint Louis, MO, USA) dissolved in 330 μL DMF was added and stirred for 90 min. The above-mentioned oleylamine-coated iron oxide nanoparticles (8 to 10 mg of Fe^3+^ contained in 8 mL) were then added to the dopamine mixture and kept at room temperature with stirring conditions for 5 hrs. Hexane (Sigma-Aldrich, Saint Louis, MO, USA) was then added to the reaction mixture to make the final volume of 100 mL, which allowed for particles to settle down in the hexane medium. The precipitate was washed with hexane and air dried for 10–20 min. The pellet was resuspended in milliQ water, and the mixture was centrifuged at 800× *g* for 20 min to remove larger aggregates. The resultant supernatant was then centrifuged again at 800× *g* for 40 min with 10 kd nano-sep (Pall Pvt Ltd., Mumbai, India) to remove any unreacted PEGs. The final concentration of the SPION was 5 mg/mL.

### 2.3. Covalent Attachment of Drug to Particles

The SPIONs were suspended in 2-(N-morpholino) ethanesulfonic acid (MES) buffer (0.1 (M), pH 4.5) (Thermo Fisher Scientific, Waltham, MA, USA). 40 mM EDC (Thermo Fisher Scientific, Waltham, MA, USA) was then added to 4 mL of SPION suspension and stirred for 10 min. This was followed by an addition of 10 mM NHS (Thermo Fisher Scientific, Waltham, MA, USA). Both the EDC and NHS were dissolved in MES buffer. The mixture was agitated for 15 min. The solution was centrifuged at 800× *g* for 40 min to remove excess EDC and NHS. Abciximab (ReoPro, Parchem Chemicals, New York, NY, USA) was added to the particle suspension and stirred overnight. The final concentration of ReoPro was 2 mg/mL. The mixture was centrifuged at 800× *g* in nano sep (100 kd) for 40 min to remove excess ReoPro, whereafter ReoPro conjugated SPION (Tx@ReoPro) was suspended in PBS buffer (pH~7.3). 

### 2.4. Optical Measurement

The absorbance spectrum of the SPION solution was measured using Thermo Scientific Evolution 300 UV-VIS (Waltham, MA, USA). The hydrodynamic diameter and zeta potential of SPIONs and Tx@ReoPro were measured using Malvern NanoZS [3].

### 2.5. Transmission Electron Microscopy of Nanoparticles

Transmission electron microscopy (TEM) was performed at 75 kV for the characterisation of the size and morphology of the nanomaterial. A drop of SPION and Tx@ReoPro solution was placed on a 300 mesh copper grid coated with carbon and dried at room temperature prior to imaging using TEM (FEI Tecnai G2 Galadriel, Hillsboro, OR, USA).

### 2.6. Fourier Transform Infra-Red Spectroscopy (FTIR) for Conjugated Nanoparticles

PEG-coated SPIONs and Tx@ReoPro were lyophilised (12 Pa, −45 °C) in a lyophiliser (TOKYO RIKAKIKAI CO., LTD, Tokyo, Japan). Approximately 3–4 mg of the amorphous samples were mixed with ~200 mg of Potassium Bromide (KBr) in a mortar pestle. The mixture was pelleted out by applying ~12-ton pressure, and finally, the FT-IR spectrum of nano-conjugants was recorded in transmission mode (Thermo Scientific, Nicolet-6700, Waltham, MA, USA). Each spectrum was recorded after 256 scans and 4 cm^−1^ wave number resolution at room temperature.

### 2.7. Magnetic Force Microscopy

Magnetic force microscopic images of SPION and Tx@ReoPro nanoparticles were obtained with the Veeco VI Innova model (Bruker Axs Pvt. Ltd., Mumbai, India) using MESP probes (Co/Cr coated). The frequency range used was from 70–80 kHz; the spring constant(k) was 2.8 N/m; the radius of curvature was 20 nm at 0.5 Hz scan rate with 256 × 256 resolution.

### 2.8. Light Transmission Aggregometry

For platelet function studies using light transmission aggregometry, 9.5 mL blood was mixed in 3.2% sodium citrate anticoagulant (final ratio 9:1 whole blood/citrate). Platelet-rich plasma (PRP) was obtained after centrifuging blood at 200× *g* for 10 min. PPP (platelet-poor plasma) was obtained by centrifugation of blood at 1500× *g* for 10 min and served as respective blank. The PRP was incubated with PEG-coated SPIONs and Tx@ReoPro (with respective concentrations, mentioned in the figure legend) at 37 °C for 15 min. Platelet aggregation was initiated by ~6 μM ADP and recorded using a Chronolog optical aggregometer (model 700, Havertown, PA, USA).

### 2.9. Whole Blood Impedance Aggregometry

Whole blood platelet aggregation was studied using a Multiplate^®^ analyser (Roche Diagnostic GmbH, Mannheim, Germany). For this study, venous blood from volunteers was collected in hirudin tubes and allowed to rest for 30 min. Blood was then incubated with PEG-coated SPIONs and Tx@ReoPro (from 1.61 μg/mL to 32.25 μg/mL) at room temperature for 10 min. By that time, 300 μL NaCl was added to the disposable test cells placed in the Multiplate^®^ instrument. Then, 300 μL of hirudinised blood treated with PEG-coated SPIONs and Tx@ReoPro was added to the test cell and incubated for 3 min at 37 °C under stirring conditions. After that, 20 μL of ADP (Multiplate^®^ ADPtest, final concentration 6.5 μM) was added to initiate platelet aggregation. The aggregations were monitored for 6 min. All the reagents used in this experiment were from Roche Diagnostic GmbH, Mannheim, Germany.

### 2.10. Free Oscillation Rheometry (FOR)

Viscoelastic whole blood coagulation measurements were performed by Free Oscillation Rheometry—FOR (ReoRox G2, MediRox AB, Nyköping, Sweden). For this study, citrated blood was incubated with PEG-coated SPIONs and Tx@ReoPro (final concentration 80 μg/mL) at room temperature for 20 min. A 50 μL ReoTRAP reagent (from ReoRox kit) and 25 μL 0.5 M CaCl_2_ (from ReoRox Kit) were mixed with 1 mL of that treated blood using a disposable 1 mL syringe by gently pipetting up and down. The blood (1 mL) was then added to the reaction chamber, whereby the measurement of viscosity and elasticity was started automatically through the ReoRox G2 software [15].

### 2.11. Thromboelastography (TEG)

Analysis of whole blood coagulation was performed by the addition of 1 mL whole blood to TEG kaolin cuvettes (Haemoscope, Skokie, IL, USA). The sample was mixed, and 2 aliquots of 450 μL each of blood were added to Tx@ReoPro (final concentration 80 μg/mL) and PEG-coated SPION (equivalent amount of Tx@ReoPro) separately. From each mixture, 360 μL of the treated blood was added to the TEG assay cups. Then, TEG tracings were recorded to follow the whole blood coagulation process. 

### 2.12. Flow Cytometry 

For flow cytometry experiments, hirudinised whole blood was incubated with varying concentrations of Tx@ReoPro (1.16 μg/mL, 8.06 μg/mL and 32.25 μg/mL) at room temperature for 10 min. After this incubation, 5 μL of treated blood was incubated with 5 μL of FITC conjugated α-fibrinogen antibody (Diapensia HB, Linkoping, Sweden, diluted 1:10) in 50 μL final volume of HEPES buffer (137 mM NaCl, 2.7 mM KCl, 1 mM MgCl_2_, 5.6 mM glucose, 1 g/L BSA, 20 mM 4-(2-Hydroxyethyl) piperazine-1-ethanesulfonic acid (HEPES), pH 7.4). The platelets were activated with 10 μM ADP at room temperature for 10 min. The activation was stopped using 600 μL HEPES buffer, and the samples were analysed by flow cytometry (Gallios; Beckman Coulter, Brea, CA, USA). For the negative control, HEPES with 10 mM EDTA was used instead of HEPES.

### 2.13. In Vitro Magnetic Resonance Imaging

In a phantom experiment, the relaxation rates R2 [1/T2, s^−1^] and R2* [1/T2*, s^−1^] were measured in vials with platelets with increasing concentrations of Tx@ReoPro (0, 0.01, 0.02, 0.03, 0.04 mg/mL). Images were acquired with a clinical Philipsachieva1.5T. The R2 was measured using a turbo spin echo sequence with 10 echoes, TR 1 sec, flip angle 90°, and a 180° inversion pulse with a first TE of 20 ms and a delta TE of 28 ms. The R2* was measured using a gradient echo sequence with a 32 echo readout, 2 sec TR, and a flip angle of 30° with a first TE of 2.8 ms and a delta TE of 2.4 ms. The region of interest was manually placed in the centre of each vial, and the relaxation rates were measured by fitting the time series to a mono-exponential signal model using the lsqcurvefit in Matlab R2017a software (The MathWorks, Inc. Natick, MA, USA). To minimise the effect of stimulated echoes in the spin echo sequence, the first echo was removed when measuring R2.

## 3. Results

### 3.1. Fabrication and Characterisation of Tx@ReoPro 

The fabrication strategy for developing the Tx@ReoPro theranostic nanosystem comprises the following steps: (i) oleylamine conjugated SPIONs synthesis by thermal decomposition; (ii) substitution of oleylamine with dopamine by ligand exchange; (iii) Polyethylene glycol (PEG) functionalisation through dopamine; (iv) zero-length EDC/NHS cross-linking to antiplatelet drug Abciximab (Repro) conjugation (as summarised in Figure 1). The advantage of the thermal decomposition method is that it can produce mono-dispersed SPIONs with a narrow size distribution, as observed in TEM and DLS. Moreover, a large number of particles can be synthesised through this method in a single batch. SPIONs were rationally chosen for this study owing to their ease in synthesis, surface modification, high colloidal stability at variable pH and temperature [16], high biocompatibility, and image-contrast ability. [17,18]. 

The homogeneity and hydrodynamic properties (zeta potential and hydrodynamic diameter) of the nanosystem were confirmed by dynamic light scattering (DLS) and transmission electron microscopy (TEM). The size distribution and mono-dispersity of the nanoparticles were evident from the TEM and poly disparity index (PDI < 0.5) obtained from the DLS study. The hydrodynamic diameter of the core particle (SPION-PEG) was 21 nm (Figure 1E), and the naked dry size was about 5 nm, as observed from TEM images (Figure 1C). As expected, the hydrodynamic diameter increased from 21 nm to 28 nm after ReoPro conjugation, and TEM images also reflected a similar trend (Figure 1D–E). The loading of ReoPro through the amide conjugation was confirmed by Fourier-transform infrared spectroscopy (FTIR). A clear shift in the spectral region, corresponding to the primary and secondary amide bond, was observed after conjugation (Appendix A). The FTIR spectra of SPION and Tx@RePro exhibit characteristic fingerprint bands positioned at 3384 and 2073 cm^−1^, which are recognised as prominent regions for SPIONs. The signature vibrations associated with the PEG can be observed at 2883 cm^−1^ (C–H asymmetric stretching), 1106 cm^−1^ (C–O–C vibration), and 958–960 cm^−1^ (CH2 rocking), confirming PEGylation. A noticeable change in the IR spectra for SPIONs and Tx@RePro in the region 3440 cm^−1^ indicates the involvement of a primary amide bond for the tethering process of the loaded drug. The absorbance spectrum of PEGylated SPIONs appeared between 200 nm and 300 nm (Figure 1F). 

### 3.2. Functional Validation of Tx@ReoPro

For functional confirmation, we employed two clinically used methods, (i) impedance-based platelet aggregometry, a whole blood-based method and (ii) light transmission platelet aggregometry, a platelet-rich plasma-based method to study the inhibition of platelet aggregation in response to Tx@ReoPro. ADP was chosen as an agonist to activate the platelets as ADP is used in routine testing of platelet function and is a natural agonist that plays a crucial role in platelet aggregation. 

In the impedance-based method, whole blood was used to study platelet aggregation. Upon agonist addition, activated platelets aggregate on the microelectrodes immersed in a cuvette of diluted whole blood (as shown in Figure 2A). The aggregation increases the impedance between the electrodes and is proportional to the intensity of the aggregation. Here, we have used a commercially available whole blood aggregometer, where two sets of electrodes are used per test. The idea is to repeat the test in identical experimental conditions (i.e., within the same cuvette). Therefore, each test always indicated two curves (blue and red) for each of the electrode pairs. This impedance-based whole blood platelet aggregometry (Figure 2A) showed that Tx@ReoPro could efficiently inhibit platelet aggregation in a dose-dependent manner from low to high drug concentrations (Figure 2B,E for whole blood aggregometry and Appendix A for light transmission aggregometry). To confirm the retention of drug function after conjugation, we have checked for an equivalent ReoPro dose (3.22 μg/mL) of free drug, and Tx@ReoPro showed a similar inhibition effect on platelet aggregation (Figure 2C). Furthermore, PEGylated SPIONs did not show any effect on platelets, which rules out the probability of false positive results (Figure 2D). Therefore, the results not only confirm that Tx@ReoPro effectively inhibited platelet aggregation but also showed no interference to the drug function after the conjugation.

To note, conventional light transmission aggregometry is based on the optical measurement of platelet aggregation in platelet-rich plasma (PRP), unlike the impedance-based aggregometry that uses whole blood. The colloidal suspension of platelets in PRP, upon aggregation, settles down, and the supernatant becomes clearer. The amount of light transmission through the suspension is measured as a percentage (%) of platelet aggregation. Therefore, increased light transmission indicates higher aggregation (Appendix A). 

Light transmission aggregometry also showed similar inhibition for Tx@ReoPro and free drug ReoPro, and bare core SPIONs had no effect on the platelet aggregation (Appendix A).

### 3.3. Inhibition of Thrombus Growth by Tx@ReoPro

Two rheometry-based assays, thromboelastography (TEG) and Free Oscillation Rheometry (FOR), have been used in this study to detect and monitor real-time thrombus formation [19,20] (as shown in Figure 3). This assay helps measure viscoelastic properties (clot size, elasticity, time, etc.) of blood clotting in laboratory conditions. TEG measures the rheometric properties of the blood clot through a pin that is suspended into a cup (37 °C) from a torsion wire, which is then connected to a mechanical–electrical transducer. The physical property of the clot is measured through the motor-driven rotation of the cup. On the other hand, in FOR, clot formation takes place in a gold-coated sample cup with a gold-coated cylinder (Bob) suspended in the centre. FOR measures the property of the clot after setting the cup into free oscillation. Therefore, this method is very sensitive, even for a very weak clot. 

A marked reduction in TEG maximum amplitude (MA), which correlates to platelet function, reflecting fibrin and platelet bonding via GPIIb/IIIa, was observed with 80 μg/mL of Tx@ReoPro treatment (Figure 3A). This concentration of Tx@ReoPro was shown to give complete inhibition of aggregation using light transmission platelet aggregometry (see Appendix A). Therefore, we used the same dose in TEG and FOR. 

Tx@ReoPro treatment also exhibited longer activated clotting time (8.8 min), along with slightly decreased kinetics (α angle, 52.9°), compared to control, though α angle was within the normal reference range.

The contribution of platelets in clot formation was more sensitively detected by FOR [21]. The Clot Onset Time 1 (COT1), i.e., the time to start clot formation when the initial strands of fibrin are formed, and Clot Onset Time 2 (COT2), i.e., the time when clot elasticity starts building up, are two important parameters in FOR to describe clot formation. For FOR, maximum clot strength G’max is comparable with TEG maximum amplitude (MA). In the presence of Tx@ReoPro (80 μg/mL), G’max was remarkably reduced, and the COT times extended (Figure 3B). Both confirm the inhibitory effect in real time. As a control, in both cases, only SPIONs did not show any remarkable effect on thrombus growth.

### 3.4. Tx@ReoPro Targeting and Theranostic Potential of Tx@ReoPro

To confirm the binding of Tx@ReoPro to the fibrinogen receptor of platelets, the ligand-based targeting was estimated using a fibrinogen binding assay (Figure 4A). Flow cytometric data showed that in the presence of Tx@ReoPro, the fluorescence intensity of FITC-conjugated α-fibrinogen antibody diminishes. This is due to the specific binding of the drug with the fibrinogen receptor GPIIbIIIa on platelets (Figure 4A). The GPIIbIIIa is an integrin receptor on the platelet outer surface. Upon platelet activation, downstream signalling activates this receptor, and as a result, fibrinogen can bind to this receptor and promote platelet aggregation. Tx@ReoPro clearly showed a shift in population due to its specific binding with the fibrinogen receptor GPIIbIIIa on platelets (Figure 4A). SPIONs alone did not show any inhibitory effect on platelet aggregation (Appendix A). Similar doses of the drug with and without conjugation showed no differences (Appendix A). 

The magnetic property of the iron oxide nanoparticle was evaluated using SPIONs and Tx@RepPro particles. Prominent magnetic force lines were observed during magnetic force microscopy (MFM), which was a clear indicator of the magnetic property of the synthesised nanoparticles (Figure 4B and Appendix A). The magnetic property of Tx@ReoPro was reflected in magnetic resonance imaging (MRI), where an increase in the dose of Tx@ReoPro showed increased R2* intensity in MRI images (Figure 4C). This increase in the dose of Tx@ReoPro also showed inhibition of platelet aggregation, as revealed by bright field microscopy (Figure 4D) for the same samples as used for MRI. 

## 4. Discussion

Platelets play a major role in arterial thrombosis and contribute to its severity. Real-time monitoring of arterial thrombosis, along with its growth restriction by targeting platelets, could, therefore, be a very interesting area of exploration for future applications in the clinical field. 

In this feasibility study, we have shown a novel theranostic nanotool, Tx@ReoPro, that can efficiently bind to platelets and restrict platelet aggregation and in vitro thrombus growth. Synthesised PEGylated SPIONs were small in size, with uniform diameter, and could easily be conjugated with the antiplatelet drug ReoPro (Figure 1). The PEGylated SPIONs did not show any effect on platelet aggregation or fibrinogen binding to platelets (Figure 2, Appendix A), indicating that the core module does not affect platelet function. The PEG, an FDA-approved drug ligand [22], not only served as a second protective bio-layer on the nanoparticle but also expanded functional space on the surface to accommodate higher numbers of drug molecules per particle. The model drug that has been used here is ReoPro, which is a monoclonal antibody that blocks the platelet fibrinogen receptor GPIIbIIIa. As a result, fibrinogen cannot bind to GPIIbIIIa receptors on the platelet’s surface, resulting in inhibition of platelet aggregation and further clot formation [15,23]. ReoPro had been in clinical use in patients with myocardial infarction or unstable angina who went for percutaneous coronary intervention (PCI). ReoPro administration was reported to reduce major adverse cardiac events (MACE) among such patients [24]. We have observed that Tx@ReoPro was highly stable and could be in aqueous suspension (PBS) for around 6 months. The functional validation of the developed nano-system-based theranostic module was performed using clinically relevant state-of-the-art methods. To confirm the functional activity of the drug after conjugation with the nanoparticle, platelet aggregometry was conducted using both impedance and light transmission-based methods (Figure 2, Appendix A). Results showed that Tx@ReoPro inhibits platelet aggregation in a dose-dependent manner (Figure 2B,E and Appendix A). A complete inhibition of aggregation was observed in light transmission platelet aggregometry at a dose of 80 μg/mL Tx@ReoPro, whereas for impedance-based whole-blood platelet aggregometry it was 32 μg/mL of Tx@ReoPro. As the blood sample processing and detection principles were different for these different aggregometry, the effective inhibitory concentrations were also found to be different. 

How inhibition of platelet function by Tx@ReoPro affects the viscoelasticity during clot formation is an important area of concern. In our study, we used TEG and FOR to show that in the presence of Tx@ReoPro, clot strength or elasticity decreased remarkably (Figure 3). To elaborate further, a decrease in the MA in the TEG experiment was also found with Tx@ReoPro. MA represents the maximal strength of the fibrin clot and is dependent on fibrin–platelet interactions via GPIIbIIIa, where the influence of platelet function is 80% and the influence of the fibrin network is 20% [19]. Therefore, ReoPro-induced inhibition in fibrinogen binding to platelets is perfectly reflected in the decrease in MA. Furthermore, G’max in the FOR system showed a significant reduction, which further proved the active functionality of Tx@ReoPro. This confirms that the conjugated Tx@ReoPro nanotool can reduce clot strength, which was the prime aim of this study. Specific inhibition of GPIIbIIIa is important to study to confirm that target (GpIIbIIIa) specific binding of ReoPro is conserved after nano-conjugation. Flow cytometric data showed that Tx@ReoPro can inhibit fibrinogen binding to activated platelets and proves that Tx@ReoPro binds specifically to GpIIbIIIa (Figure 4).

From the above discussion, it is evident that in an in vitro condition, Tx@ReoPro can efficiently inhibit platelet function and clot formation by target-specific inhibition of the platelet fibrinogen receptor, GPIIbIIIa. Along with the therapeutic aspects (platelet function and clot inhibition), Tx@ReoPro has the potential to aid in diagnosis/monitoring with prompt identification of the site of thrombus formation, making it a potential theranostic nanotool for arterial thrombosis. MFM and MRI images have shown that Tx@ReoPro is a magnetically active molecule and that the magnetic properties were retained even after conjugation (Figure 4 and Appendix A). The MRI results, along with respective bright field microscopy images, confirmed that Tx@ReoPro could be used for thrombus detection through MRI and could restrict platelet aggregation and, therefore, potentially reduce thrombus build-up (Figure 4).

In a clinical set-up during PCI, the patients for abciximab therapy received intravenous abciximab as a bolus dose of 0.25 mg/kg body weight (~20 mg to start) followed by a continuous infusion of 0.125 gm/kg/min up to a maximal dose of 10 μg/min for 12 h, amounting to 7.2 mg within a timeframe of 12 h with obvious individual variation [25]. Therefore, to check the comparable amount of drug, in this study, Tx@ReoPro was used from 1 to 80 μg/mL to see its antiplatelet effect.

## 5. Conclusions

We have performed an in vitro feasibility study to develop proof-of-concept results on a self-reporting theranostic tool for Arterial Thrombosis. The synthesised Tx@ReoPro can be considered a potential theranostic nanotool that can perform the dual activity of thrombus detection and thrombus growth restriction by inhibiting platelet aggregation that can be effectively detected by MRI.

## Figures and Tables

**Figure 1 bioengineering-10-01020-f001:**
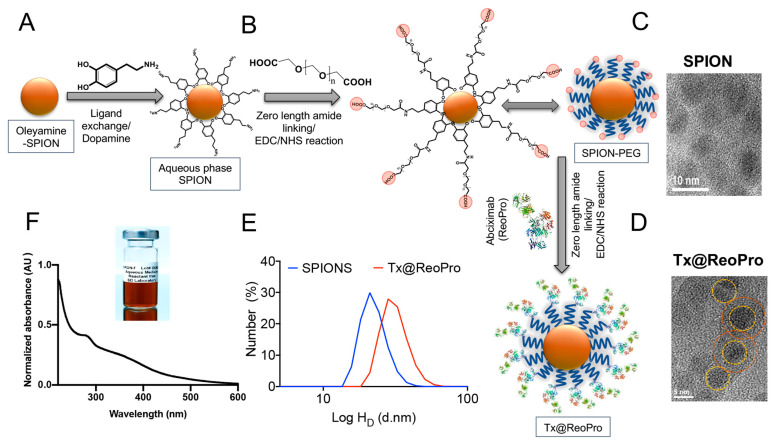
Chemical fabrication of Tx@ReoPro nanotool and its physicochemical characterisation: (**A**) SPION conjugated with oleylamine; (**B**) formation of aqueous phase SPION by substituting oleylamine with dopamine through ligand exchange method; (**C**) PEG functionalisation of the dopamine-conjugated SPION (SPION-PEG) is ~5 nm diameter in TEM micrograph image (scale bar 10 nm); (**D**) Abciximab (ReoPro) drug amalgamation to the chemically modified SPION through amide bond by EDC/NHS reaction resulted in Tx@ReoPro. TEM micrographs of Tx@ReoPro is showing a diameter of around 7 nm (scale bar 5 nm); (**E**) Hydrodynamic diameter of SPION-PEG (21.04 nm) is represented by blue line, and hydrodynamic diameter of Tx@ReoPro (28.21 nm) is represented by red line. (**F**) Shows absorption spectra of the SPION nanocomposite Tx@ReoPro.

**Figure 2 bioengineering-10-01020-f002:**
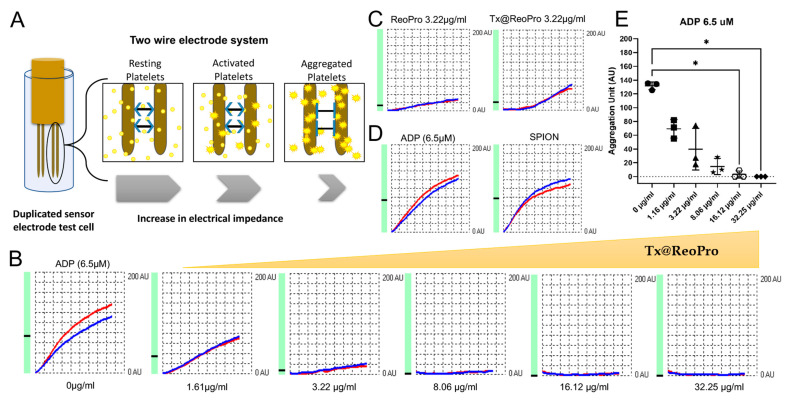
Inhibition of platelet aggregation by Tx@ReoPro: (**A**) Schematic representation of whole blood impedance-based platelet aggregometry method; (**B**) ADP (6.5 μM)-induced platelet aggregation inhibition in presence of increasing concentration of Tx@ReoPro; (**C**) Effect of ReoPro and Tx@ReoPro on 6.5 μM ADP induced activated platelets where drug concentration is same (i.e., 3.22 μg/mL); (**D**) Effect of SPION on platelet aggregation with ADP (6.5 μM); (**E**) One-way ANOVA analysis of Aggregation Unit (final aggregation) showed a significant change in platelet aggregation in presence of Tx@ReoPro (16.12 μg/mL and 32.25 μg/mL) when activated with ADP 6.5 μM, where * = *p* value < 0.05, and *n* = 3. Blue and red indicate duplicate test for each electrode pair (**B**–**D**).

**Figure 3 bioengineering-10-01020-f003:**
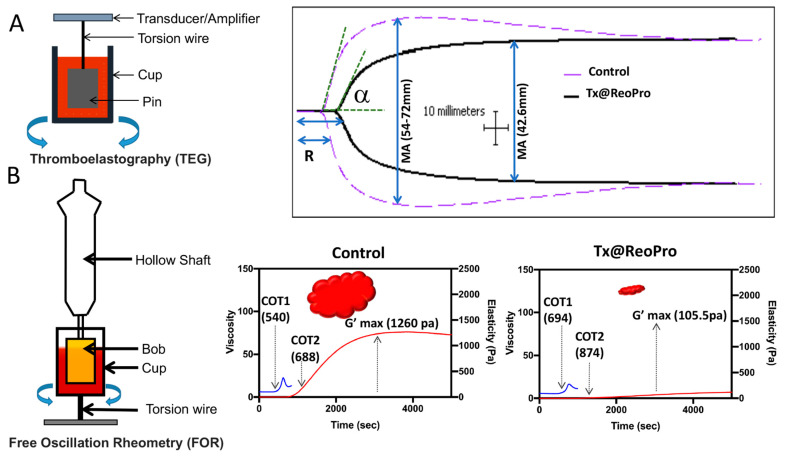
Rheometric analysis of thrombus formation in the presence of Tx@ReoPro (80 μg/mL) using TEG and FOR: (**A**) Thromboelastography (TEG)-based real-time thrombus monitoring showed reduced maximum amplitude (MA) (solid black line), i.e., 42.6 mm in presence of Tx@ReoPro compared to control set (dotted purple line); (**B**) Free Oscillation Rheometry (FOR) showed longer clot onset times (COT1 representing blue line and COT2 representing red line) with reduced clot strength (G’max), i.e., 105.5 pa in presence of Tx@ReoPro compared to control set. Each of these experiments was performed in duplicate set and, in total, 4 times. One set of characteristic TEG and FOR data is presented here.

**Figure 4 bioengineering-10-01020-f004:**
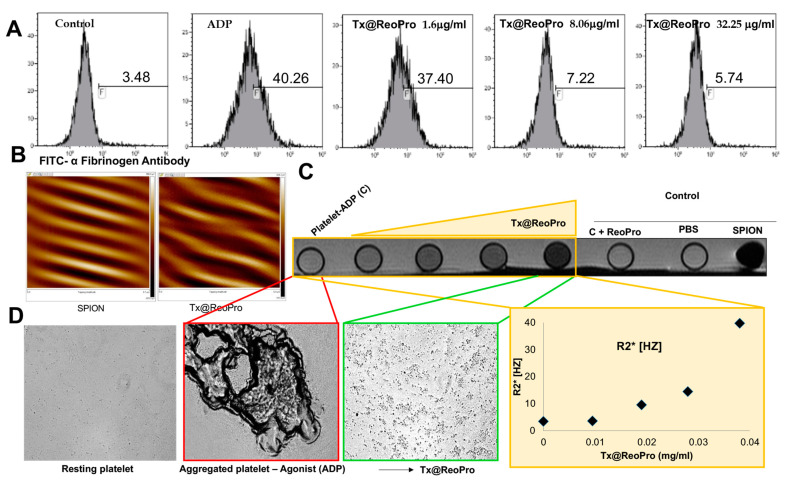
Inhibition of the fibrinogen receptor GPIIbIIIa by Tx@ReoPro along with its real-time monitoring: (**A**) Fibrinogen binding experiments were performed using flow cytometry, where FITC-conjugated fibrinogen was used to monitor Tx@ReoPro-mediated inhibition of GPIIbIIIa when activated with ADP (10 μM); F is representing percentage of population; (**B**) MFM images showed prominent magnetic force lines, indicating the active magnetic property of both SPION and Tx@ReoPro; (**C**) MRI shows increased R2* intensity with increasing dose of Tx@ReoPro; (**D**) Represents bright-field microscopic image of the same samples (resting platelets, platelets activated with ADP 10 μM with and without Tx@ReoPro) from MRI experiment. The flow cytometry was performed twice, and PoC MRI experiment was performed once.

## Data Availability

The data are available on request and appropriate justification.

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
