# Peer review of "Self-Reporting Theranostic: Nano Tool for Arterial Thrombosis"

_bioengineering, 2023, doi:10.3390/bioengineering10091020_

Round 1

Reviewer 1 Report

The paper by S. Deb et al provides the results of in vitro studies for innovative theranostic platform Tx@RePro, i.e. ReoPro conjugated magnetic nanoparticles, combining anti-platelet activity of Abciximab with functionalities of superparamagnetic nanoparticles. 

I have only minor comments: 

1. In the introduction  section the abbreviations introduced in abstract (AT, SPIONS) should be explicated again since the abstract is a metadata of the paper, not the part of the text.

2. Line 40, CT abbreviature is not explicated 

3. Line 258: "showed extend of inhibition of platelet aggregation" - please, check the grammar: the verb"extend" is used as a noun here.

4. Figure 4  caption, line 342: "MFM images showed prominent magnetic force lines indicating the active magnetic property of both SPION and Tx@ReoPro". The aim of scanning probe microscopy is to visualize nanostructures, not the force lines.  Since MFM shows  a quasi-domain structure, we probably have some kind of SPIONS magnetic aggregation here. It is desirable to provide the topographic images corresponding to these scans in the main text or Supplementary section.

Author Response

Reviewer 1:

The paper by S. Deb et al provides the results of in vitro studies for innovative theranostic platform Tx@RePro, i.e. ReoPro conjugated magnetic nanoparticles, combining anti-platelet activity of Abciximab with functionalities of superparamagnetic nanoparticles. 

Reply: We thank the reviewer for kind evaluation and providing constructive feedback on our MS. We are also glad that the reviewer finds this in vitro feasibility study as ‘innovative theranostic platform’. We have further improve the MS to address all the comments and highlighted the changes in yellow.

I have only minor comments: 

  1. In the introduction  section the abbreviations introduced in abstract (AT, SPIONS) should be explicated again since the abstract is a metadata of the paper, not the part of the text.

Reply: Thank you for pointing this out. We have revised and modified.

  1. Line 40, CT abbreviature is not explicated 

Reply: Thank you for pointing this out. We have revised and modified.

  1. Line 258: "showed extend of inhibition of platelet aggregation" - please, check the grammar: the verb"extend" is used as a noun here.

Reply: We have revised and modified the sentence.

  1. Figure 4  caption, line 342: "MFM images showed prominent magnetic force lines indicating the active magnetic property of both SPION and Tx@ReoPro". The aim of scanning probe microscopy is to visualize nanostructures, not the force lines.  Since MFM shows  a quasi-domain structure, we probably have some kind of SPIONS magnetic aggregation here. It is desirable to provide the topographic images corresponding to these scans in the main text or Supplementary section.

Reply: Thank you for your comment. You have rightly pointed out that MFM shows  a quasi-domain structure, and we probably have SPIONS magnetic aggregation here. In our opinion, that is very much possible as the MFM is also performed in a dry lyophilized state. Therefore, self-aggregation is possible at the dry state. Furthermore, they are magnetic and hence self-aggregation due to magnetic field is highly probable at the dry stage. As per your suggestion, we have now provided the topographic images corresponding to these scans (Supplementary Figure 4) and as explained, the size will not be matched exactly as in DLS due to their dry and assembled state.

Summary: As per the reviewer’s suggestion we have explained all the abbreviation in the revised MS. We have thoroughly revised and adjusted the text. A topographic MFM image corresponding to figure 4 has been included in the supplementary section.

Reviewer 2 Report

In their manuscript, Deb et al. report the development and the initial efficacy evaluation of iron oxide-based nanoparticles functionalized with Abciximab, a monoclonal antibody against platelet fibrinogen receptors GPIIb/IIIa. Although the topic of the paper is potentially interesting to the audience, the data seem very preliminary and the conclusions far-fetched. Additional experiments and analyses are necessary before the manuscript can be published.

Major comments:

1. The authors should provide a clear distinction concerning the potential indication of the used drug. As such, the readers are led to believe that ReoPro can be used in patients with arterial thrombosis (including stroke). In fact, ReoPro is currently discontinued, and was only indicated only for prevention of thrombosis in patients undergoing e.g. PCI.  It is not suitable for thrombus monitoring or thrombolysis and was withdrawn from phase 3 trials in patients with stroke due to the safety reasons.

2. Drug concentration used in the synthesis process, as well as drug binding efficacy and the end-concentration/mg Fe must be given. The authors should compare it to the recommended dose in patients.

3. Statistical analysis missing completely. Was there only one sample used? How many blood donors were investigated?

4. Evaluation of particle toxicity is missing. It is absolutely indispensable to provide the cell viability (e.g. human blood cells) data upon incubation with different concentrations of control particles and their drug-loaded counterparts for at least 48h.

5. It is not clear why the authors used ADP and not thrombin for aggregometry analyses. ADP is not an ideal stimulus and very large inter-donor differences in platelet responsiveness to ADP were reported.  

Further, why was only one concentration of ReoPro-particles tested in TEG and FOR? There is no explanation given, and the selected concentration (80 µg/ml) was much higher that the effective concentrations tested in aggregometry (3.22 µg/ml).

6. The statement that the developed nanosystem allows real-time drug status monitoring by MRI is not supported by the experimental data.  The time course of thrombus formation was not investigated. All that the authors did was to confirm that NPs can be visualized in concentration-dependent manner by MRI (which is true for all iron-oxide based nanoparticles). However, the detectability of NP signal does not indicate the thrombus status!

7. The authors did not test biocompatibility, but claim in the Discussion that their NP are biocompatible. Further, they state that the particles are stable, but no data concerning particle characterization upon storage are shown.

8. Conclusions contain statements that grossly exceed the results. Both in the last paragraph of Introduction and in Conclusions, the authors forgot to mention that all their data are done only in vitro and (presumably) on a limited number of subjects.

The information is missing that the particles may be suitable for prevention of thrombus, but not for thrombolysis /management of existing thrombus.

 Minor comments:

1. English needs editing.

Author Response

In their manuscript, Deb et al. report the development and the initial efficacy evaluation of iron oxide-based nanoparticles functionalized with Abciximab, a monoclonal antibody against platelet fibrinogen receptors GPIIb/IIIa. Although the topic of the paper is potentially interesting to the audience, the data seem very preliminary and the conclusions far-fetched. Additional experiments and analyses are necessary before the manuscript can be published.

Reply: Firstly, the authors want to thank the reviewer for considering the work as ‘potentially interesting’. We are also grateful for the critical evaluation. As the reviewer mentioned, we would like to clarify again that this is an in vitro feasibility study to develop proof-of-concept results. Therefore, an animal study was out of the scope at the current stage. Now we are planning to do so, as we have obtained the confirmatory data set to go for an animal study with different drugs and combinations of nanosystems. This has now been clarified in the revised manuscript.

Major comments:

  1. The authors should provide a clear distinction concerning the potential indication of the used drug. As such, the readers are led to believe that ReoPro can be used in patients with arterial thrombosis (including stroke). In fact, ReoPro is currently discontinued, and was only indicated only for prevention of thrombosis in patients undergoing e.g. PCI.  It is not suitable for thrombus monitoring or thrombolysis and was withdrawn from phase 3 trials in patients with stroke due to the safety reasons.

Reply: Thank you for pointing this. As mentioned above, we have used this as a model drug for this feasibility study, other drugs can be easily adapted and conjugated to the platform. To elaborate, we have used Reo-Pro as a model drug (as you have also mentioned in it’s use in PCI) to show how a MRI visible SPION conjugated drug can be used in combined diagnosis and therapeutic (theranostic) purpose. The concept can further invigorate in this field on real time monitoring of effect of anti-platelet drugs. We have clarified the rationale behind choosing this drug in introduction and discussion part of the revised MS, highlighted yellow.

  1. Drug concentration used in the synthesis process, as well as drug binding efficacy and the end-concentration/mg Fe must be given. The authors should compare it to the recommended dose in patients.

Reply: The drug concentration used during the synthesis was 1mg/ml. The final concentration of the SPION was 5mg/ml in Tx@ReoPro. We have revised the MS and highlighted.

During the optimization, the major concern was to check the efficacy in terms of function and the contrasting ability of the platform to allow non-invasive monitoring. However, we have shown clearly that the nano-system and the bare drug were similar in function (Figure 2C, Supplementary Figure 3, Supplementary Figure 5). Flow cytometric data has shown how the Tx@ReoPro can bind to its platelet specific glycoprotein receptor GpIIbIIIa in a dose dependent manner and comparable to the bare drug concentration (Figure 4a, Supplementary Figure 5). This also prove that even after conjugation with the drug the nanosystem did not lose its functional activity and binding affinity towards platelets. 

  1. Statistical analysis missing completely. Was there only one sample used? How many blood donors were investigated?

Reply: Thank you for the comment. For most of our experiment we have performed 2-4 times as a standard practice. The Tx@ReoPro has also shown a clear dose dependent platelet function inhibition for aggregometry (optical and whole blood) and flow cytometry experiments. Platelet function inhibition by Tx@ReoPro was cross-checked and found similar to its respective drug (without carrier) concentration. We further showed that SPION itself did not show any such effect on platelet function.

  1. Evaluation of particle toxicity is missing. It is absolutely indispensable to provide the cell viability (e.g. human blood cells) data upon incubation with different concentrations of control particles and their drug-loaded counterparts for at least 48h.

Reply: Thank you for your concern. We have used 2 clinically relevant components here namely ReoPro (already explained in previous reply) and SPION. We have rationally chosen iron oxide nanoparticle (SPION) because of their safety index and clinical use. FDA already approved several iron oxide nanoparticles based system such as Venofer®, Ferrlecit®, INFed®, Dexferrum®, and Feraheme® etc to name a few. A comprehensive overview of all FDA approved systems can be found here: https://link.springer.com/article/10.1007/s40005-017-0370-4/tables/4.

Therefore, no extended acute and chronic toxicity at the cellular level been measured. Furthermore, for platelets, due to their shorter lifetime outside the native environment (4 hours only), an 48h incubation will not show platelet activity and/function.

  1. It is not clear why the authors used ADP and not thrombin for aggregometry analyses. ADP is not an ideal stimulus and very large inter-donor differences in platelet responsiveness to ADP were reported.  

Reply: Thank you for this technical comment. We choose ADP over thrombin because we wanted to work in systems as close to the in vivo situation as possible. And as thrombin also causes coagulation, we could not use this agonist without making additional modifications to the test systems which might lead to platelet pre-activation and less reliable results. In addition, ADP is an important secondary feedback activator in all activation pathways of platelets (including that of thrombin, as it is normally released from dense granule) and is important to stabilize platelet aggregates. ReoPro inhibits platelet GpIIbIIIa receptor, which is the fibrinogen binding receptor. Upon stimulation, both ADP and thrombin can activate this receptor and promote platelet aggregation. Therefore, either of these can be used to see the inhibitory effect of Tx@ReoPro.

Further, why was only one concentration of ReoPro-particles tested in TEG and FOR? There is no explanation given, and the selected concentration (80 µg/ml) was much higher that the effective concentrations tested in aggregometry (3.22 µg/ml).

Reply: We are sorry for the confusion. In the revised version we have explained and clarified to better understand. We have measured platelet function in 2 different types of aggregometry namely light transmission aggregometry (using PRP) and impedance-based aggregometry (using whole blood). Therefore, different doses have been used to study the dose dependency.

A complete inhibition of aggregation was observed in light transmission platelet aggregometry at a dose of 80 µg/ml. We have then used the same dose to study TEG and FOR.

Reply: Whereas impedance-based whole blood platelet aggregometry showed complete inhibition of platelet around 32 µg/ml.  To confirm the retention of drug function after conjugation, we have checked an equivalent ReoPro dose (3.22 µg/mL) of free drug and Tx@ReoPro which showed similar inhibition effect on platelet aggregation (Figure 2C).

In summary, the results showed (Figure 2 and supplementary Figure S2B and S3) that Tx@ReoPro effectively inhibited platelet aggregation and retained its drug function after the conjugation.

  1. The statement that the developed nanosystem allows real-time drug status monitoring by MRI is not supported by the experimental data.  The time course of thrombus formation was not investigated. All that the authors did was to confirm that NPs can be visualized in concentration-dependent manner by MRI (which is true for all iron-oxide based nanoparticles). However, the detectability of NP signal does not indicate the thrombus status!

Reply: To clarify ‘real time’, we wanted to state that the Tx@ReoPro can be detected through MRI while it is inhibiting platelet function. In Figure 4C, it was shown that, Tx@ReoPro can be detected through MRI while it was inhibiting ADP activated platelets (cross-checked with light microscopy in Figure 4D). Therefore, it is evident that Tx@ReoPro can be detected in real time in an in-vitro scenario as well as can inhibit platelet aggregation.

  1. The authors did not test biocompatibility, but claim in the Discussion that their NP are biocompatible. Further, they state that the particles are stable, but no data concerning particle characterization upon storage are shown.

Reply: As mentioned above, we have used only clinically relevant and approved materials and therefore, a full standing biocompatibility study is not performed. Particle characterization included in the MS (Figure 1). To add, have observed that in a 4°C storage, PEGylated iron oxide nanoparticles are stable for at least 6 months.

  1. Conclusions contain statements that grossly exceed the results. Both in the last paragraph of Introduction and in Conclusions, the authors forgot to mention that all their data are done only in vitro and (presumably) on a limited number of subjects.

Reply: We have revised the conclusion as suggested.

The information is missing that the particles may be suitable for prevention of thrombus, but not for thrombolysis /management of existing thrombus.

Reply: In this feasibility study with in vitro set up thrombolysis /management of existing thrombus condition was out of scope. However, in our future study, we are going to address thrombolysis.

 Minor comments:

  1. English needs editing.

Reply: Thank you. Revised

Summary:

The reviewer has rightly pointed out that ‘ReoPro is currently discontinued’ and the authors are also in agree with reviewer. We have used ReoPro as a targeting molecule and a model drug to check whether and how a SPION conjugated drug can be used in theragnosis to small aggregates. We have clearly mention the reasons for selecting ReoPro as a model drug system. With the findings that we are reporting, we are convinced that a study of real time monitoring in an in vivo model is very much needed and a necessary next step.

Reviewer 3 Report

The idea presented in this manuscript is interesting, but alas it is described in a rather casual and inaccurate manner, which prevents the acceptance of it. In particular, the manuscript somewhat looks as a poorly assembled patchwork, where clear and accurate paragraphs are mixed with others, very poor in language at the point that they are hardly understandable. In particular, the description of nanoparticle preparation is so confusing that it is quite useless for the reader willing to repeat the experiments. Moreover, some essential information is lacking (i.e. some quantities) or obviously absurd (i.e. unacceptable weights of some reagents such as 200g of PEG). Finally, PEG is not really PEG but instead PEG dicarboxylic acid...

Here below I have reported a brief list of mistakes/typos that contribute to substantially lower the whole quality of the manuscript, which is anyway unacceptable.

Line 79 benzyle

L 80 oleyamine

L 84 precipitant

L 95 olyamine

L 99 precipitation

L 127 mortar pestle

L 170-171 Incomprehensible sentence

Author Response

Reviewer 3:

The idea presented in this manuscript is interesting, but alas it is described in a rather casual and inaccurate manner, which prevents the acceptance of it. In particular, the manuscript somewhat looks as a poorly assembled patchwork, where clear and accurate paragraphs are mixed with others, very poor in language at the point that they are hardly understandable.

Reply: The authors want to thank the reviewers for considering the work as interesting. We have thoroughly revised the MS including language, continuity and comprehensiveness of the MS.

In particular, the description of nanoparticle preparation is so confusing that it is quite useless for the reader willing to repeat the experiments. Moreover, some essential information is lacking (i.e. some quantities) or obviously absurd (i.e. unacceptable weights of some reagents such as 200g of PEG). Finally, PEG is not really PEG but instead PEG dicarboxylic acid...

Reply: We have updated and rewrite the method part specially the nanoparticle synthesis segment in the revised MS.

Here below I have reported a brief list of mistakes/typos that contribute to substantially lower the whole quality of the manuscript, which is anyway unacceptable.

Line 79 benzyle

Reply: Revised

L 80 oleyamine

Reply: Revised

L 84 precipitant

Reply: Revised

L 95 olyamine

Reply: Revised

L 99 precipitation

Reply: Revised

L 127 mortar pestle

Reply: Revised

L 170-171 Incomprehensible sentence

Reply: Revised

Round 2

Reviewer 2 Report

The authors provided several explanations in response to the comments and made changes in the manuscript to include the indication of the model drug, provide better description of methods and reduce the exaggerated statements in Conclusions.  However, these changes do not reflect the effort of performing a “major revision” recommended by this Reviewer.  

The authors MUST provide additional experiments and analyses before the manuscript is publishable.

Major comments:

1.The authors did not test biocompatibility, but claim in the Introduction and Discussion that their NP are biocompatible, based on the fact that OTHER iron oxide formulations were approved by the FDA.  This is as convincing as stating that “broccoli are vegetables and are green, therefore carrots – which are also vegetables – are green, too”.

This also indicates, that the authors have no knowledge whatsoever about the requirements for particle safety, as every slightest modification results in a new type of particles, which must be independently evaluated! As their particles are intended for intravascular therapy, the authors must provide the basic hemocompatibility (hemolysis, particle stability in blood) and biocompatibility data (e.g. leukocyte viability upon incubation with different concentrations of control particles and their drug-loaded counterparts for at least 48h).

2. Statistical analysis is still missing completely. The authors did not answer how many blood donors were investigated. In Fig. 2B no time of experiment is given. The authors must provide additional panel showing a dose-dependent % of decrease in platelet aggregation at the end of the measurement and compare the study groups statistically.

3. The description of methods are still incomplete, ending with “aggregation was initiated by 6µM ADP” but the aggregation monitoring time is not given.

4. In the results, the authors still have not mentioned that ”impedance-based whole blood platelet aggregometry showed complete inhibition of platelet around 32 µg/ml”. Instead, 3,22 µg/mL are mentioned as effective dose.  Also, in the revised text, there is still no explanation given that the concentration selected in TEG and FOR (80 µg/ml) was based on the light transmission aggregometry results, which are only shown in supplement. The authors should be aware that their explanations to the reviewer will not be seen by the readers and that they must include appropriate descriptions in the text.   

5. Further, the authors must address in the Discussion the differences between the effective dose in whole blood aggregometry (3,22 µg/mL) versus the effective dose in light transmission aggregometry (80 µg/mL).

6. The authors must also discuss their results of effective drug concentrations in vitro in comparison to the recommended dose in patients.

Minor comments:

1. English still needs editing.

Author Response

Enclosed please find the reply.
